# CaMKIIα as a Promising Drug Target for Ischemic Grey Matter

**DOI:** 10.3390/brainsci12121639

**Published:** 2022-11-29

**Authors:** Nane Griem-Krey, Andrew N. Clarkson, Petrine Wellendorph

**Affiliations:** 1Department of Drug Design and Pharmacology, University of Copenhagen, 2100 Copenhagen, Denmark; 2Department of Anatomy, Brain Health Research Centre and Brain Research New Zealand, University of Otago, Dunedin 9016, New Zealand

**Keywords:** CaMKII, excitotoxicity, GHB analogues, glutamate, HOCPCA, ischemia, neuroprotection, stroke

## Abstract

Ca^2+^/calmodulin-dependent protein kinase II (CaMKII) is a major mediator of Ca^2+^-dependent signaling pathways in various cell types throughout the body. Its neuronal isoform CaMKIIα (*alpha*) centrally integrates physiological but also pathological glutamate signals directly downstream of glutamate receptors and has thus emerged as a target for ischemic stroke. Previous studies provided evidence for the involvement of CaMKII activity in ischemic cell death by showing that CaMKII inhibition affords substantial neuroprotection. However, broad inhibition of this central kinase is challenging because various essential physiological processes like synaptic plasticity rely on intact CaMKII regulation. Thus, specific strategies for targeting CaMKII after ischemia are warranted which would ideally only interfere with pathological activity of CaMKII. This review highlights recent advances in the understanding of how ischemia affects CaMKII and how pathospecific pharmacological targeting of CaMKII signaling could be achieved. Specifically, we discuss direct targeting of CaMKII kinase activity with peptide inhibitors versus indirect targeting of the association (*hub*) domain of CaMKIIα with analogues of γ-hydroxybutyrate (GHB) as a potential way to achieve more specific pharmacological modulation of CaMKII activity after ischemia.

## 1. Introduction

Stroke is the third-leading cause of death and disability worldwide and its burden is expected to increase with an aging population [1]. Today, there are 6.5 million deaths from strokes every year, and over 50% of the surviving stroke patients suffer from permanent disabilities, causing an enormous public health burden [1,2]. There are two main types of strokes, ischemic and hemorrhagic, whereof ischemic stroke is the most prevalent one [1,3]. An ischemic stroke happens when the blood flow to a certain brain region is compromised by an obstruction of a blood vessel. The reduction in blood supply leads to energy failure of cells within the affected area as well as the induction of various signaling cascades and inflammation processes, ultimately resulting in cell death and infarct formation [4,5]. In particular, signaling anomalies of the main excitatory neurotransmitter glutamate (*excitotoxicity*) play a substantial role in mediating ischemic cell death [6,7].

Despite the heavy disease burden of ischemic stroke, treatment options are limited. To stop the progression of irreversible damage to neighboring tissue over time, timely restoration of blood flow has proven as an effective therapy capable of salvaging ischemic brain tissue [8]. This can be achieved by pharmacological thrombolysis with recombinant human tissue-type plasminogen activator (tPA, alteplase) or by mechanical thrombectomy [9,10]. However, there is a high number of patients who do not meet the criteria for recanalization therapy, and even the majority of patients receiving the therapy still experience permanent disabilities [11,12,13]. In fact, the narrow time window of 4.5 h after symptom onset and the risk of hemorrhagic transformation results in as little as ~5% of patients actually receiving tPA [11,14]. Collectively, this highlights the critical need for novel therapies improving the long-term outcome for patients. Tremendous efforts have been made to develop therapies targeting underlying cell death mechanisms to save dying tissue acutely after the initial injury (*neuroprotectants*), or therapies aiming to augment the function of surviving tissue and promote functional recovery after stroke [15,16]. Failures in the design of both preclinical and clinical studies have, however, among other reasons, hampered translation into the clinic [17]. Yet, a recent phase III trial (ESCAPE-NA1) on the peptide nerinetide has recently brought new hope to the field by demonstrating that neuroprotection could be a feasible treatment option [18]. Nerinetide inhibits the protein-protein interaction of *N*-methyl-D-aspartate (NMDA)-type glutamate receptors and the scaffold protein post-synaptic density protein 95 (PSD-95), thereby uncoupling NMDA receptors from excitotoxic signaling pathways [18,19]. Although the study observed drug-drug interactions of nerinetide with tPA, it did demonstrate that targeting a key signaling pathway upstream of excitotoxic cell death can result in clinically beneficial outcomes in the patient group not receiving tPA [18].

This review will focus on another potential target for neuroprotection, which also directly interacts with NMDA receptors, namely Ca^2+^/calmodulin-dependent protein kinase II alpha (CaMKIIα). CaMKIIα is a major mediator of both physiological and pathological glutamate signaling [20,21,22,23], and post-insult CaMKII inhibition has been shown to substantially reduce cell death in vitro and in vivo [24,25,26]. However, as different subtypes of CaMKII are expressed in most cell types, fulfilling critical signaling tasks, targeting CaMKIIα after ischemia requires strategies that are both subtype-selective as well as specific for pathological regulatory pathways governed by this important kinase. Here, we briefly introduce CaMKII and its regulation with specific focus on emerging functional roles of its hub domain. Next, we address the role of CaMKII in physiological glutamate signaling and present how ischemia induces dysregulation of CaMKIIα. Furthermore, we discuss recent advances in pharmacological intervention with CaMKIIα. In particular, we address the differential outcomes by interfering directly with CaMKII kinase activity versus indirect modulation of CaMKII function via targeting of its hub domain.

## 2. CaMKII

The CaMKII protein family is encoded by four separate genes yielding four different variants (CaMKIIα, β, γ and δ). Albeit expressed throughout the whole body, CaMKII is most known for its neuronal and cardiac functions. Specifically, CaMKIIδ is highly expressed in cardiomyocytes contributing to cardiac placemaking [23] and calcium handling [27], whereas CaMKIIα and CaMKIIβ are predominantly expressed in neurons accounting for a remarkable 1–2% of total brain protein [28]. The latter emphasizes the importance of CaMKIIα/β as a signaling module, which was shown to mediate synaptic plasticity underlying learning and memory [20,29,30,31]. Of note, this review will focus on neuronal CaMKIIα and CaMKIIβ, and the term CaMKII will thus refer to CaMKIIα/β if not otherwise specified.

### 2.1. CaMKII Structure and Activity Regulation

Each CaMKII protein comprises four domains, i.e., a N-terminal Ser/Thr-specific kinase domain connected to a regulatory segment and a variable linker region, which is followed by a C-terminal association (*hub*) domain responsible for giving the enzyme a unique architecture (Figure 1A, reviewed in [32]). In brief, CaMKII monomers can assemble into large oligomeric structures (*holoenzyme*), typically made up of 12 or 14 subunits, where the hub domains form two donut-shaped rings stacked on top of each other, arranging the kinase domains around it (Figure 1A) [32,33,34]. The activity of CaMKII is controlled by intracellular Ca^2+^ levels. Under basal conditions, the regulatory segment of each subunit binds to its own kinase domain and thereby blocks the central substrate binding site and thus suppresses its catalytic activity to a minimum (*autoinhibition*). Increases in intracellular Ca^2+^ levels activate the enzyme by Ca^2+^/calmodulin (CaM) binding to the regulatory segment. Since the binding site for Ca^2+^/CaM overlaps with the region responsible for autoinhibition, Ca^2+^/CaM binding relieves the inhibition and exposes the access to the substrate site (*stimulated activity*, Figure 1B) [35,36]. Holoenzyme activity is further regulated by multiple threonine residues within the regulatory segment. As such, Ca^2+^/CaM binding and subsequent release of autoinhibition exposes Thr286 (CaMKIIα numbering; used throughout the text if not otherwise stated; Thr287 in other subtypes), which is centrally located within the autoinhibitory region of the regulatory segment. Thr286 can be phosphorylated by a neighboring activated subunits (*trans autophosphorylation*), which prevents the interaction of the regulatory segment with the kinase domain and thus keeps the kinase active also beyond the initial Ca^2+^ signal when Ca^2+^ levels return to basal [37]. This state is referred to as *autonomous* activity or Ca^2+^-independent activity [36,38,39]. CaMKII requires Ca^2+^/CaM binding and autophosphorylated Thr286 (pThr286) for full catalytic activity. In contrast, autonomous activity after Ca^2+^/CaM dissociation amounts to only ~15–40% of its maximum activity [40,41]. Thr286 autophosphorylation enhances the affinity of Ca^2+^/CaM for CaMKII (*CaM trapping*), thus reducing the rate of Ca^2+^/CaM dissociation [42]. Moreover, when Ca^2+^/CaM is not bound, CaMKII can be autophosphorylated at two sites located within the CaM binding domain, i.e., Thr305/306. Phosphorylation at these sites can occur in an intra-subunit reaction (*cis*) and inhibits Ca^2+^/CaM binding, making CaMKII insensitive to intracellular Ca^2+^ signals (Figure 1B) [32,43]. Furthermore, other autophosphorylation sites on CaMKII have been described [44,45], yet their functional consequence are less well characterized. For instance, it has been shown that CaMKII can be autophosphorylated at Thr253 in vitro and in vivo, which, in contrast to Thr286 and Thr305/6, does not directly influence kinase activity, but instead influences the interaction of CaMKII with specific binding proteins, thereby affecting the subcellular targeting of the holoenzyme [45]. Of note, CaMKII is further regulated by phosphatase activity, and distinct autophosphorylation sites have been shown to undergo dephosphorylation at different rates [46,47].

The aforementioned holoenzyme structure and activation properties are conserved among all four CaMKII proteins. In fact, they share ~95% sequence identity in the kinase and regulatory domain as well as ~80% in the hub domain [32]. Notably, CaMKII is also subjected to alternative splicing generating additional diversity with ∼40 variants [48,49,50]. The four CaMKII subtypes and its splice variants differ the most in sequence and length of the linker domain [32], which is suggested to contribute to subtype-specific functional characteristics. For example, the identity of the linker influences the activation properties of CaMKII [34,46], which was demonstrated by showing that a CaMKIIα variant without its linker domain is less sensitive to activation by Ca^2+^/CaM compared to a CaMKIIα variant with 30-residue linker [34]. In addition, linker identity was shown to affect subcellular targeting of the holoenzyme. As such, the linker from CaMKIIβ, but not CaMKIIα, contains an F-actin binding domain [51]. As CaMKII subtypes can assemble into homomers, but also mixed heteromers [52,53], subunit composition might determine the activation and functional properties of the CaMKII holoenzyme.

### 2.2. Emerging Functional Roles of the CaMKII Hub Domain

The hub domain is a unique feature of CaMKII, carrying a well-established role in CaMKII oligomerization thus securing overall kinase fidelity [54]. Hence, in addition to providing a scaffold for the arrangement of the kinase domains, the unique holoenzyme architecture also gives rise to functional properties, including its response to Ca^2+^/CaM [34,55]. Specifically, oligomerization provides a framework for inter-subunit autophosphorylation of Thr286 by bringing the kinase domains in close proximity and increasing the effective local concentration of subunits [56]. Thr286 autophosphorylation concomitant with CaM trapping likely contributes to cooperative activation of the holoenzyme by Ca^2+^/CaM and allows CaMKII to act as a frequency decoder by translating the frequency of cellular Ca^2+^ signals into specific amounts of kinase activity under physiological conditions [34,41,55]. The functional importance of the CaMKII holoenzyme structure is further highlighted by human patients with neurodevelopmental defects, carrying a mutation in the CaMKIIα hub domain (p.His477Tyr) shown to cause impaired oligomerization [57].

Moreover, the hub assembly was suggested to be endowed with a high degree of structural flexibility [32,33,58]. As such, in vitro experiments suggest that CaMKII activation triggers the exchange of subunits between activated/autophosphorylated but also non-activated holoenzymes. Intriguingly, it has been speculated to enable the spread of activity via autophosphorylation [58,59,60]. The proposed underlying molecular mechanism involves the interconversion of dodecameric or tetradecameric CaMKII by the release and addition of vertical dimeric subunits, suggesting the tetradecameric oligomer as an intermediate state. It has been shown that the regulatory segment can bind to the hub domain, which is believed to initiate a conformational change resulting in hub destabilization and dimer release, which can then integrate into nearby holoenzymes [58,59,60]. Although this clearly demonstrates remarkable dynamics of the holoenzyme assembly in vitro, the function of subunit exchange and relevance in vivo remains elusive [61].

Finally, kinase and hub domains have been shown to interact and to functionally influence each other, thus adding further complexity to the regulation of CaMKII activation [34,49,62]. Specifically, Sloutsky and colleagues provided structural evidence that kinase domains can form direct physical contact with the hub domain, which was suggested to contribute to CaMKII autoinhibition and/or to affect Ca^2+^/CaM sensitivity of the holoenzyme [49]. They also demonstrated that the Ca^2+^/CaM sensitivity of a holoenzyme depends on the identity of its hub domain by showing that the CaMKIIα kinase domain is more sensitive to Ca^2+^/CaM when fused to the CaMKIIβ hub domain with no linker in comparison to its combination with the CaMKIIα hub domain without linker. Importantly, these experiments show that the hub domain is capable of allosteric modulation of kinase activity [49]. Intriguingly, it was recently demonstrated that pharmacological interaction with the hub domain can functionally modulate the CaMKII holoenzyme, e.g., by affording neuroprotection (see Section 4.3 and Section 4.4) [25].

### 2.3. CaMKII in Physiological Glutamate Signaling

CaMKIIα and CaMKIIβ are reported to be expressed in a 3:1 ratio in forebrain regions, and their concentration is especially high in the dendritic spines of cortical and hippocampal neurons [28,63]. Neuronal CaMKII is most known for its crucial role in the mediation of glutamate signaling underlying learning and memory processes [41]. Specifically, upon Ca^2+^ influx through NMDA receptors CaMKII gets activated by Ca^2+^/CaM and subsequent Thr286 autophosphorylation. Autophosphorylation not only infers autonomous activity, but also affects the subcellular targeting of CaMKII [64,65]. As such, increased pThr286 levels are associated with holoenzyme translocation from the center of the dendritic spine to the post-synaptic density (PSD) [66,67]. At the PSD, CaMKII can interact with several synaptic proteins, such as the NMDA receptor subunit GluN2B [30,68,69]. Interestingly, GluN2B binding renders CaMKII autonomously active because the binding site for GluN2B on CaMKII overlaps with the binding site for its regulatory domain, thus preventing autoinhibition [64]. The co-localization with GluN2B strategically locates CaMKII in close proximity of sites of Ca^2+^ entry into spines as well as several substrates centrally involved in strengthening of synaptic connections, i.e., long-term potentiation (LTP) [36,41]. Prominent substrates of CaMKII in LTP are α-amino-3-hydroxy-5-methyl-4-isoxazolepropionic acid (AMPA)-type glutamate receptors, whose phosphorylation by CaMKII results in both augmented channel conductance and increased number of synaptic receptors [36,70,71]. The critical involvement of both CaMKIIα and CaMKIIβ in synaptic plasticity has been demonstrated by genetic knockdown of either of them leading to reduction in LTP and learning in mice [20,72]. Other genetic studies have further demonstrated that both CaMKIIα activation and GluN2B co-localization are essential for normal synaptic plasticity by showing that knock-in mice carrying mutations that impair GluN2B-CaMKIIα binding (GluN2B L1298A/R1300Q) or Thr286 autophosphorylation (CaMKIIα T286A) also show deficits in LTP [29,73]. In contrast, the activity of CaMKIIβ is not required for LTP, but CaMKIIβ instead fulfils important structural roles by promoting the targeting of CaMKIIα to the synapse which is mediated by the ability of CaMKIIβ to bind to F-actin via its linker domain [67,72]. Consequently, it has been suggested that the CaMKIIα/β heteromer contributes to structural plasticity as a consequence of synaptic activity, and it has been shown that CaMKII is implicated in the formation of dendritic spines and synapses [74,75].

Interestingly, CaMKII and Thr286 are also critically involved in long-term depression (LTD), leading to weakening of excitatory glutamatergic synapses and potentiation of inhibitory synapses [76,77]. The mediation of this apparent opposing effect on plasticity is enabled by finetuning its activity and subcellular location via inhibitory Thr305/306 autophosphorylation [78]. As such, pThr305/306 prevents GluN2B binding, reduces CaMKII targeting to the PSD and directs CaMKII towards LTD-related substrates [66,73]. This highlights a dual role of CaMKII in the mediation of LTP and LTD as well as the complexity of CaMKII signaling.

## 3. Disturbances in CaMKIIα Signaling after Ischemic Stroke

Excessive release of glutamate accompanied by overstimulation of glutamate receptors, especially NMDA receptors, is a hallmark of the pathology of ischemic stroke [6]. Yet, extensive efforts to directly target the NMDA receptor for the treatment of acute ischemic stroke could not be translated to the clinic [17,79,80]. To be able to differentiate cell death from cell survival pathways or pathological from physiological functions, it has been suggested to target key signaling pathways downstream of NMDA receptors instead [16] - a concept that has recently delivered encouraging results in a phase III clinical trial (ESCAPE-NA1) [18]. CaMKII undertakes such a key signaling role downstream of the NMDA receptor by integrating physiological but also pathological Ca^2+^ signals, and CaMKII has thus emerged as a target in ischemic stroke [21]. Here, we will briefly describe relevant pathological signaling pathways after ischemia and we will discuss the involvement of CaMKII in ischemic cell death. For a detailed description of the pathology of ischemic stroke the reader is referred to [4,5,7]. Of note, this review focuses on neuronal CaMKII signaling in grey matter pathology. In contrast, the entire role of CaMKII as a target in white matter remains to be investigated.

### 3.1. Pathophysiology of Ischemic Stroke

The pathophysiology of ischemic stroke is complex and involves the activation of several interconnected signaling cascades cumulating in cell death and tissue damage (Figure 2) [4]. The area close to the occluded artery and most affected by hypoperfusion is called the *ischemic core*, which usually experiences irreversible tissue damage within minutes after injury [4]. The blood flow in the tissue surrounding the infarct core is less severely affected, mostly due to residual perfusion from collateral blood vessels. Here, structural and metabolic integrity is maintained, but the tissue is characterized by functional impairments. This area is called the *ischemic penumbra* [81], and it is the target for therapeutic interventions, like restoration of blood flow or experimental neuroprotective therapies [17]. Yet, if no treatment is initiated, potentially viable penumbral tissue gradually gets integrated into the infarct core [82].

Deprivation from glucose and oxygen initially results in energy failure of neurons in the ischemic core. Specifically, neurons are unable to produce adenosine 5′-triphosphate (ATP), which leads to the impairment of energy-depended processes and disruption of ion homeostasis (Figure 2). As a consequence, neurons depolarize excessively, triggering the release of toxic concentrations of excitatory neurotransmitters, which fail to be taken up [5]. In fact, accumulation of glutamate in the extracellular space plays a central role in the mediation of ischemic cell death by initiating prolonged stimulation of glutamate receptors, which in turn leads to the excessive influx of Ca^2+^, Na^+^ and H_2_O (*excitotoxicity*). Whereas Na^+^ influx is mainly associated with the formation of cytotoxic oedema, intracellular Ca^2+^ overload triggers various cytotoxic signaling pathways, such as the dysregulation of Ca^2+^-dependent enzymes including CaMKII (Figure 2) [4,21]. Notably, various enzymes (proteases, lipases or nucleases) get overactivated as a consequence of Ca^2+^ overload, which are responsible for diminishing cellular integrity [4]. For instance, a major contributor to excitotoxic cell death is calpain, a Ca^2+^-dependent protease, which alters the stability but also function of several proteins critically involved in cellular survival by proteolytic processing [83].

The initial period after vascular occlusion is termed the acute phase (<3 days). Of note, in addition to the described neuronal changes, the acute phase is characterized by interlinked pathological processes involving all cell types of the neurovascular unit. As such, ischemia also triggers inflammation, oxidative stress, apoptosis and breakdown of the blood-brain barrier (BBB), which together with excitotoxicity ultimately cumulates into infarct formation (Figure 2) [4,82]. After the acute phase, damaged tissue transitions from injury to repair in the subacute (>3 days) and chronic phase (>1 month) [84,85]. The repair phase is characterized by a period of heightened plasticity, where neurons within the area surrounding the infarct (peri-infarct) and their connected areas facilitate structural plasticity and functional remodeling. This involves axonal growth, dendritic remodeling and the formation of new synapses leading to remapping of sensorimotor functions, which is believed to underlie a limited degree of spontaneous recovery of functional deficits after stroke [15,85,86,87]. On a molecular level, these changes are mediated by similar mechanisms that underlie synaptic plasticity during learning and memory, such as LTP and dendritic remodeling [88,89,90].

### 3.2. CaMKII Dysregulation after Ischemia

Several studies suggest the involvement and dysregulation of CaMKII in cell death in the acute phase after ischemia [21,91], which is also the focus of this review. Interestingly and despite its major role in synaptic plasticity [36], CaMKII signaling has, to our knowledge not been investigated during the repair phase after stroke.

#### 3.2.1. Increased Autophosphorylation

Acute CaMKII inhibition has been shown to protect neurons from excitotoxic cell death, clearly highlighting a role of CaMKII signaling in ischemic cell death [21,24,26,92]. However, at this point it is unclear exactly how CaMKII mediates ischemic cell death. Several mechanisms for its dysregulation have been described, and the majority of studies are in agreement with the following overall sequence of events. Initially, intracellular Ca^2+^ overload introduced by excitotoxic glutamate signaling results in rapid activation of CaMKII via Ca^2+^/CaM binding. Subsequently, CaMKII undergoes increased and sustained autophosphorylation at Thr286 after ischemia in comparison to sham non-stroke controls, rendering CaMKII autonomously active. Moreover, CaMKII gets persistently translocated to the PSD, where it co-localizes with the NMDA receptor subunit GluN2B. Here, CaMKII is centrally located in close proximity of a number of substrates implicated in both cellular survival and excitotoxic cell death, and ischemia-induced hyperactivation of CaMKII is believed to lead to excessive substrate phosphorylation (Figure 3) [21,91].

Specifically, multiple in vivo models of ischemic injury could demonstrate persistent translocation of CaMKII to the PSD concomitant with elevated pThr286 levels in this fraction, while pThr286 levels and total CaMKII expression were decreased in the cytosolic fraction [26,93,94,95,96,97,98]. This pattern of CaMKII dysregulation has for instance been shown by Matsumoto and colleagues in the ischemic cortex following 1 and 2 h of ischemia as well as following 2 h of reperfusion in a mouse model of focal ischemic stroke (transient middle cerebral artery occlusion, tMCAO) [95]. In addition, rapid increase in autophosphorylation at Thr253 was observed after tMCAO [94], and similar autophosphorylation patterns for both Thr286 and Thr253 could also be demonstrated after global ischemia [26,97,98,99], which shares important pathophysiological mechanisms with ischemic stroke. However, in contrast to focal ischemic injury, global ischemia occurs after transient loss of oxygen supply to the whole brain, for example after cardiac arrest, and is characterized by neuronal death in brain regions most susceptible to ischemic injury, such as the hippocampus and striatum [100,101].

#### 3.2.2. Self-Association 

In contrast to the apparent increase in Thr286 autophosphorylation after ischemia, it has been shown that the catalytic activity of CaMKII is considerably decreased in the membrane and cytosolic fraction after both global and focal ischemia [93,102,103,104,105,106]. It has been suggested that the CaMKII inactivation after ischemia is mediated by vast self-association of multiple CaMKII holoenzymes (*aggregation*). As such, CaMKII aggregation is favored in vitro after Ca^2+^/CaM activation under conditions mimicking ischemia (low ATP, high adenosine diphosphate (ADP) and low pH) and involves the interaction of a kinase domain with the regulatory domain of a neighboring holoenzyme, which results in reduced catalytic activity [107,108,109,110]. Consequently, CaMKII aggregation has been proposed as a natural protective mechanism to minimize excessive kinase activity after ischemia, yet the potential neuroprotective effect of aggregation requires further studies [21]. However, the fact that pThr286 levels are increased after ischemia clearly indicates that at least a subpopulation of total CaMKII is hyperactive and is likely contributing to ischemic cell death, since inhibition of CaMKII activity has been shown to be neuroprotective [24,26].

#### 3.2.3. GluN2B Co-Localization

Co-localization of CaMKIIα with GluN2B has been shown to be crucially involved in the mediation of ischemic cell death since mice carrying a mutation in the binding site for CaMKIIα on GluN2B (L1298A/R1300Q) are protected from global ischemia [111]. Interestingly, it appears that the generation of autonomous activity and/or the strategic positioning of the holoenzyme at the PSD as a consequence of CaMKIIα-GluN2B co-localization could be the decisive factors for the neurotoxic role of this interaction as opposed to CaMKII-mediated phosphorylation of GluN2B at Ser1303. This is based on the fact that the level of Ser1303 phosphorylation of GluN2B was not changed in the hippocampus after global or in cortex after focal ischemia, thus indicating that it is not involved in ischemic cell death [94,112].

#### 3.2.4. Calpain-Mediated Proteolytic Processing

Recently, another pathological regulatory mechanism of CaMKII was described, which could contribute to the hyperactivity of a subpopulation of CaMKII after ischemia. Specifically, CaMKII was found to be proteolytically processed by the protease calpain after ischemia, which generates a stable kinase fragment with uncontrolled kinase activity (∆CaMKII, 31 kDa) [34,113]. As the cleavage site was mapped to the beginning of the regulatory domain (i.e., M^281^↓H^282^), cleavage of CaMKII at this site generates a N-terminal fragment consisting of the kinase domain only (Figure 4). In fact, ∆CaMKII is missing the regulatory motif responsible for kinase autoinhibition and activity regulation by Ca^2+^/CaM, making this fragment constitutively active [34,70,114,115]. Although proteolytic processing of CaMKII by calpain could already be described in vitro in 1989 [116,117,118], its relevance in vivo was only recently addressed. Intriguingly, ∆CaMKII has been detected exclusively after ischemic stroke (photothrombotic stroke) [113]. Furthermore, in vitro experiments showed that proteolytic processing of CaMKII at this site requires prior Ca^2+^-stimulation and Thr286 autophosphorylation [113], resembling ischemic conditions of CaMKII [26]. Proteolytic processing of CaMKII might thus represent a pathological mechanism downstream of ischemia-activated and Thr286 phosphorylated CaMKII, and the uncontrolled kinase activity of ∆CaMKII might contribute to the neurotoxic role of sustained CaMKII activity after ischemia by phosphorylating specific neuronal proteins to direct cell death. Although it is well-established that calpain overactivation contributes to excitotoxic cell death [119,120,121], the neurotoxic role of ∆CaMKII requires further experimental validation.

#### 3.2.5. Potential Downstream Effectors 

Collectively, ischemia might differently affect individual subpopulations of CaMKII. As such, one CaMKII pool might be inactivated by inter-holoenzyme aggregation, whereas another pool might undergo sustained activation involving increased autophosphorylation, translocation to the PSD as well as proteolytic activation. Currently it is unclear which one of these pathological mechanisms, leading to aberrant CaMKII activity, are most centrally involved in the mediation of excitotoxic cell death. Furthermore, it is unknown which downstream pathways of CaMKII are affected. Yet, a variety of substrates of CaMKII have been suggested as potential downstream effectors contributing to cell death (reviewed in [21]). For instance, CaMKII can phosphorylate the GluA1 subunit of AMPA receptors at Ser831, which mediates increased channel conductance for Ca^2+^ and may thus enhance excitotoxicity [122]. Notably, increased Ser831 phosphorylation after ischemia has been demonstrated in vivo [94,123]. CaMKII was also shown to phosphorylate the GABA_B1_ subunit of GABA_B_ receptors at Ser867, which leads to enhanced GABA_B_ receptor internalization [124]. Interestingly, phosphorylation of Ser867 was increased after sustained activation of glutamate receptors in cultured neurons [124], potentially contributing to downregulation of GABA_B_ receptors and consequently reduced inhibition in the acute phase after ischemia [125]. In addition, CaMKII has been shown to modulate signaling of several other proteins with a previously described role in ischemic cell death [21], such as L-type voltage-dependent Ca^2+^ channels [126], connexin hemichannels [127] and acid-sensing ion channels [128].

In contrast, other downstream effects of CaMKII could mediate cell survival instead [21]. As such, phosphorylation of neuronal nitric oxide synthase (nNOS) by CaMKII results in reduced production of neurotoxic nitric oxide [129]. Furthermore, multiple studies have associated CaMKII with the activation of the transcription factor cAMP response element—binding protein (CREB), which is a well-described regulator of neuronal survival [130,131,132]. A potential role of CaMKII signaling in cell survival is supported by the fact that CaMKIIα knock-out (*Camk2a^−/−^*) mice display increased vulnerability to ischemic injury after tMCAO and photothrombotic stroke, as evidenced by increased cortical infarct sizes compared to wildtype (*Camk2a^+/+^*) littermates [25,133]. Correspondingly, prolonged CaMKII inhibition in cortical primary neurons was shown to enhance cell death in response to an excitotoxic insult [92]. Together, this led to the hypothesis that a sustained loss of CaMKII activity increases neuronal susceptibility to ischemic injury, whereas acute short-term CaMKII inhibition after injury is neuroprotective [21,92]. However, a recent study by Rumian and colleagues could not reproduce the sensitizing effect of long-term CaMKII inhibition in vitro, and they further showed that *Camk2a^−/−^* mice are protected from ischemic injury in the hippocampus after global ischemia [134]. Although discrepancies between these studies could be explained by technical differences such as divergent inhibitor concentrations or expressional changes in the employed *Camk2a^−/−^* mouse lines [134], this clearly highlights the complexity of CaMKII signaling after ischemia. Simultaneously this indicates that cell and/or injury type might determine which downstream pathway of CaMKII will dominate. In line, tissue specific responses of CaMKII to ischemia were shown by Skelding and colleagues, who found differential rates of Thr253 autophosphorylation between striatum and cortex after tMCAO [94].

Further studies are needed to elucidate the downstream signaling of CaMKII after ischemia, which would ideally require more specific genetic approaches, such as conditional gene knock-out, to overcome compensatory mechanisms, or developmental effects using germline knock-out models. Moreover, most studies evaluating downstream effects of CaMKII have been performed with inhibitors lacking subtype selectivity and/or showing off-target effects (CaMKII inhibitors will be addressed below). Consequently, better inhibitors would be desirable to study CaMKIIα signaling more specifically.

## 4. Targeting CaMKIIα after Ischemia

### 4.1. CaMKII Inhibitors 

The most widely used CaMKII inhibitors are the small molecule KN93 and the peptide CN21 [135,136]. KN93 was developed in 1991 and was commonly believed to interact directly with CaMKII to prevent kinase activation by competing with Ca^2+^/CaM [135]. Yet, a recent study showed that KN93 inhibits CaMKII activation by directly binding to Ca^2+^/CaM instead [137]. In accordance with reported cellular effects, KN93 indeed only inhibits the stimulated and not the autonomous kinase activity, and its binding to Ca^2+^/CaM might also explain several off-target effects of the compound. For instance, Ca^2+^/calmodulin-dependent protein kinase IV (CaMKIV), another member of the CaM-kinase family, as well as voltage-gated potassium and calcium channels have been shown to be inhibited by KN93 [138,139,140]. In contrast, CN21 is a peptide derived from the natural and specific CaMKII inhibitor protein CaM-KIIN, which was fused to the cell penetrating peptide tat, thus enabling cellular and brain permeability [136]. Tat-CN21 inhibits CaMKII activity by binding to the kinase domain. Specifically, it interacts with the region responsible for the binding of the regulatory domain around Thr286 when CaMKII is autoinhibited. It means that prior enzyme activation by Ca^2+^/CaM is required for tat-CN21 binding, but also that it is able to inhibit both Ca^2+^/CaM-stimulated and autonomous activity [136]. Since GluN2B interacts with an overlapping site on the CaMKIIα kinase domain, tat-CN21 interferes with the interaction of CaMKIIα with GluN2B [24,136]. Furthermore, tat-CN21 displays higher selectivity for CaMKII compared to KN93, yet it does not discriminate between CaMKII subtypes [136]. Of note, further improvements of CN21 led to peptides with increased potency, such as CN19o (IC_50_ for CN21 0.1 μM [136] and IC_50_ for CN19o 0.4 nM determined for the substrate syntide2 at CaMKIIα) [141]. Moreover, as an alternative to tat-conjugated peptides, myristoylated CaM-KIIN (myr-CN27) has been used [142]. These peptides are collectively referred to as CN-peptides.

### 4.2. The Kinase Domain as a Target for Neuroprotection

CN-peptides are neuroprotective in vitro and in vivo (Table 1) [24,26,143]. Specifically, tat-CN21 could mediate substantial neuroprotection as evidenced in an infarct size reduction of 70% when treatment was initiated 1 h after artery occlusion at the same time as reperfusion induction in the tMCAO model [24]. Moreover, post-insult treatment with CN-peptides facilitates neuroprotection after global ischemia [26,143]. These studies were supported by in vitro findings, showing that tat-CN21 reduces glutamate-induced neuronal cell death in primary cortical and hippocampal cultures when applied 2 and 6 h after the insult, respectively [24,92]. In contrast, KN93 is only neuroprotective in vitro when present during, but not after the insult [24,92]. Since KN93 only inhibits Ca^2+^/CaM-stimulated activity, this led to the hypothesis that autonomous CaMKII activity is crucial for ischemic cell death and thus the relevant molecular species of CaMKIIα to target in neuroprotection. Although the inhibitor studies with KN93 are complicated by its off-target effects, this hypothesis was further supported by a recent study showing that CaMKIIα knock-in mice with impaired Thr286 autophosphorylation (T286A) are protected from hippocampal cell death induced by global ischemia [26]. However, the same mutation did not affect infarct sizes in mice after focal ischemic injury introduced by tMCAO [144], which again points towards tissue and/or injury specific responses of CaMKII. As such, Ca^2+^/CaM-stimulated CaMKII activity might play a bigger role than autonomous activity in mediating excitotoxic cell death after focal ischemia compared to global ischemia. Alternatively, autonomous activity could be generated by mechanisms other than Thr286 autophosphorylation or other autophosphorylation sites might be involved.

### 4.3. Ligands Targeting the CaMKIIα Hub Domain 

Recently, another class of compounds could be linked to CaMKII modulation, namely small molecules related to γ-hydroxybutyrate (GHB). GHB itself is a metabolite of γ-aminobutyric acid (GABA) and was shown to be neuroprotective in pre-clinical models of ischemic stroke in rodents [146,147]. Although the mechanism behind GHB’s neuroprotective effect is not completely understood, it likely involves two distinct binding sites within the CNS. As such, GHB is a partial agonist at GABA_B_ receptors with millimolar potency, but was also recently found to target CaMKII *alpha* with low micromolar affinity (Figure 5) [25,148]. Several series of structural analogues of GHB have been developed, which display improved selectivity and affinity for CaMKIIα in comparison to GHB. Importantly, these compounds, referred to as *GHB analogues*, are devoid of GABA_B_ receptor binding [148,149,150,151,152,153]. Furthermore, the binding site could be mapped to a deep cavity within the hub domain of CaMKIIα, thus making them not only the first subtype selective CaMKIIα ligands, but also the first ligands to interact selectively with the hub domain [25]. This striking selectivity was confirmed for the conformationally restricted and brain permeable GHB analogue 3-hydroxycyclopent-1-enecarboxylic acid (HOCPCA) exhibiting a complete lack of binding to *Camk2a^−/−^* mouse brain tissue when tested in in vitro autoradiography using its tritiated radioligand (Figure 5) [25,150,154]. Moreover, the selectivity of HOCPCA was further verified by showing that it does not bind to 45 known CNS targets [155].

### 4.4. The CaMKIIα Hub Domain as a Target for Neuroprotection

The exact functional consequences of GHB analogue binding to the CaMKIIα hub domain is not completely understood. Importantly, it was found that HOCPCA mediates neuroprotection in vitro and in vivo (Table 1) [25,145]. In fact, HOCPCA improved the functional outcome of mice after ischemic stroke with an extended treatment window in two permanent models, namely 3 h after permanent middle cerebral artery occlusion (pMCAO) and 12 h after photothrombotic stroke [25,145]. Similar neuroprotective effects were seen after early HOCPCA treatment 30 min post-stroke in a model characterized by gradual reperfusion (thromboembolic stroke) and when tested in combination with tPA [145]. Moreover, HOCPCA increased cellular survival of cultured cortical neurons after an excitotoxic insult only when applied 1 h after the insult, but had no effect when applied during the insult [25]. This is in contrast to tat-CN21, which decreases cell death when present during the excitotoxic insult [24,92], thus pointing towards differential mechanism of actions of tat-CN21 and HOCPCA. Indeed, whereas CN-peptides directly interfere with kinase activity [136], an allosteric effect on holoenzyme function is envisaged for HOCPCA based on its interaction with the hub domain [25].

#### A Longer Treatment Window via Targeting of the CaMKIIα Hub?

The late-onset effect of HOCPCA in vitro might indicate that a functional effect of HOCPCA depends on pathological dysregulation of CaMKIIα. In support, hitherto no effects on classical CaMKIIα signaling could be found under naïve nonpathological conditions. HOCPCA did not affect basal Thr286 autophosphorylation in cortical primary neurons, LTP in mouse hippocampal slices nor the catalytic activity of CaMKIIα when tested in a kinase assay (ADP-Glo) [25]. Instead, a range of GHB analogues were found to substantially increase stabilization of the purified CaMKIIα hub domain in a thermal shift assay [25,152,153], suggesting that GHB analogues might alter holoenzyme functionality by introducing a conformational change of the hub domain and thereby affecting CaMKII oligomerization or protein-protein interactions. At this point it is unclear how hub domain stabilization would translate into neuroprotection, yet it was shown that HOCPCA affects CaMKIIα biochemistry after ischemia in vivo [145]. As such, HOCPCA was able to alleviate ischemia-induced dysregulation of different subpopulation of CaMKIIα after pMCAO. Particularly, HOCPCA treatment both normalized an ischemia-induced decrease of cytosolic pThr286 after stroke and downregulated the expression of the constitutively active cleavage product ∆CaMKII [145]. Intriguingly, this shows that HOCPCA alters CaMKII activity exclusively after ischemia, which is ultimately expected to affect downstream CaMKII signaling pathways. It is tempting to speculate that a conformational change introduced by HOCPCA binding underlies both its effect on cytosolic pThr286 and proteolytic processing, yet this requires further studies. Nonetheless, these results further point towards a pathospecific mechanism of HOCPCA and highlight the therapeutic potential in modulation of the hub for achieving neuroprotection.

## 5. Concluding Remarks on Targeting CaMKIIα after Ischemia

Due to its central role in regulating physiological and pathological glutamate signaling directly downstream of NMDA receptors, CaMKIIα emerged as a promising target for neuroprotective treatments for ischemic stroke and potentially other diseases characterized by glutamate and Ca^2+^ dysregulation [21,24,26]. Such a treatment would ideally be CaMKII subtype specific and would only target pathological functions of ischemia-activated CaMKIIα to avoid the interference with crucial physiological functions of CaMKIIα. Accordingly, the therapeutic potential of CN-peptides is challenged by possible effects on learning and memory as it has been shown that tat-CN21 inhibits LTP induction [156]. Although a certain degree of learning or memory impairments might be tolerated for the acute treatment of ischemic stroke, treatment with CN-peptides might also entail cardiological side effects by inhibiting CaMKIIδ in cardiomyocytes [157]. In contrast, pharmacological modulation of the CaMKIIα hub domain by GHB analogues might offer a therapeutic strategy with increased selectivity based on HOCPCA’s lack of binding to CaMKIIβ/δ/γ and apparent pathospecific mechanism [25,145]. To this end, further studies are warranted to elucidate the exact mechanism of action of pharmacological modulation of the CaMKIIα hub domain by GHB analogues. Nonetheless, the emerging functional role of the hub domain involving its influence on kinase activity, oligomerization and potentially activation triggered subunit exchange highlight the hub domain as an interesting pharmacological target [34,49,59]. In line, as HOCPCA does not directly interfere with kinase activity [25], it would be interesting to test its effect on functional recovery when treatment is initiated during the subacute phase after stroke. In contrast to the acute phase, treatments during the subacute phase aim to enhance endogenous repair mechanisms, which are for instance involving brain plasticity mechanisms that parallel those in learning and memory and might thus depend on CaMKII signaling [86,87]. Of note, CN-peptides would likely be counter-indicated for treatment during the recovery phase due to their negative effect on LTP in contrast to HOCPCA [156].

Moreover, identification of downstream effectors and interacting proteins contributing to CaMKII subsequent to ischemic cell death could guide the design of novel treatments either by directly targeting of downstream effector proteins or their protein-protein interactions with CaMKII. This would possibly allow the targeting of a specific pathological pathway over physiological and/or cell survival functions of CaMKII. For instance, this could recently be achieved by inhibiting the interaction of CaMKIIβ with GABA_B_ receptors using a small interfering peptide derived from the CaMKII phosphorylation site on GABA_B1_ [158]. This peptide could successfully restore ischemia-mediated downregulation of GABA_B_ receptors and increased neuronal survival after glutamate insult in vitro [158]. Other intriguing strategies could be to target the interaction of calpain with CaMKII with the goal to prevent ischemia-specific proteolytic processing of CaMKII [113].

Temporal and mechanistic delineation of the ischemia-induced dysregulation of CaMKII would be helpful to guide dosing regimens of pharmaceutical interventions targeting this important brain kinase. This would ideally include different types of ischemic injury (e.g., permanent versus gradual reperfusion and grey matter versus white matter injury models) and the combination with reperfusion therapy modelling thrombectomy or drug-induced thrombolysis [17]. Furthermore, more studies are needed to investigate the contribution of CaMKII signaling to ischemic cell death in non-neuronal cells such as endothelial cells or oligodendrocytes [159] or even infiltrating immune cells. Finally, with the eventual biological validation of targeting CaMKII in ischemic stroke, a broader range of acute brain injuries might be interesting to consider.

## Figures and Tables

**Figure 1 brainsci-12-01639-f001:**
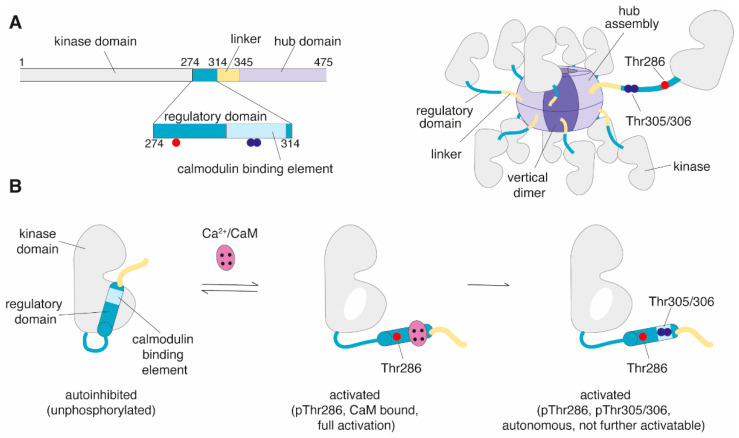
Structure and activation of Ca^2+^/calmodulin-dependent protein kinase II (CaMKII) [32,41]. (**A**) (Left) Domains of individual CaMKII subunit: kinase (grey), regulatory (teal), linker (yellow) and hub (purple) domain. The regulatory domain contains important autophosphorylation sites Thr286 (red) and Thr305/306 (blue). Thr305/306 are located within the calmodulin binding element (light blue). (Right) Holoenzyme assembly made up of 12 subunits. (**B**) Regulation of CaMKII activity. Of note, the hub domain is not shown. During basal states, the regulatory domain binds to the kinase domain and blocks access to the substrate binding site (autoinhibition). CaMKII gets activated by Ca^2+^/CaM binding to the regulatory segment, which releases autoinhibition of the kinase domain and subsequently exposes Thr286. Trans autophosphorylation of Thr286 by a neighboring subunit infers autonomous enzyme activity. CaMKII can get autophosphorylated in an intra-subunit reaction at Thr305/306 after Ca^2+^/CaM dissociation [46]. Thr305/306 autophosphorylation inhibits re-binding of Ca^2+^/CaM. Architecture is shown for CaMKIIα but overall domain structure, assembly and activation mechanism is similar for other CaMKII subtypes.

**Figure 2 brainsci-12-01639-f002:**
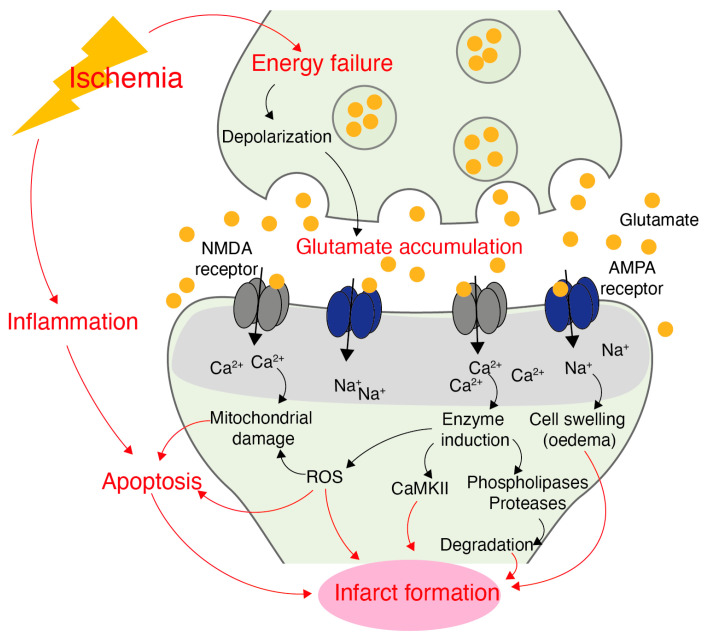
Simplified overview of excitotoxicity after ischemic stroke [4]. Obstruction of blood flow leads to energy failure and excessive depolarization of neurons. Glutamate accumulates in the synaptic cleft and overstimulates glutamate receptors, such as *N*-methyl-D-aspartate (NMDA)– and α-amino-3-hydroxy-5-methyl-4-isoxazolepropionic acid (AMPA)-type glutamate receptors. Overstimulation of glutamate receptors results in increased cell permeability and dramatic increases in Ca^2+^ and Na^+^ influx, creating an osmotic gradient resulting in cell swelling and oedema. Intracellular Ca^2+^ overload dysregulates various Ca^2+^-dependent pathways and overstimulates enzymes, including CaMKII. Overactivity of phospholipases and proteases diminishes cellular integrity, as well as overstimulation of enzymes responsible for producing excessive amounts of reactive oxygen species (ROS), which in turn damage any cellular component. Ca^2+^ overload and ROS also mediate mitochondrial damage, ultimately promoting apoptotic cell death. Excitotoxicity, oxidative stress and apoptosis lead to ischemic cell death and infarct formation in conjunction with inflammation.

**Figure 3 brainsci-12-01639-f003:**
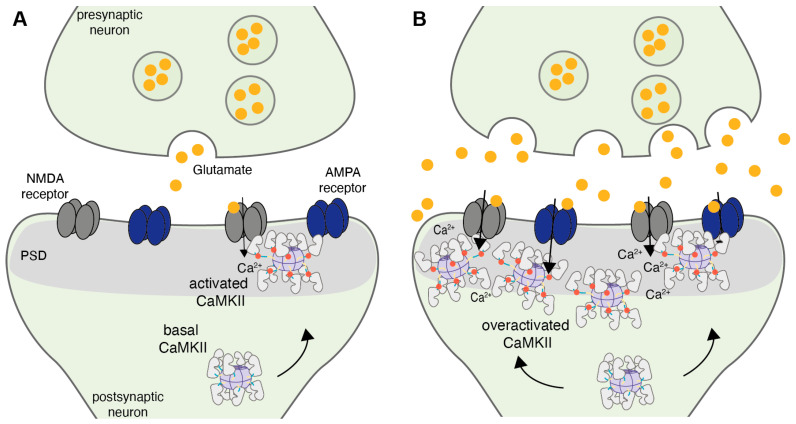
CaMKII translocation under physiological and ischemic conditions [21]. (**A**) Physiological synaptic activity at excitatory glutamatergic neurons leads to Ca^2+^ influx through NMDA receptors. Subsequent Ca^2+^/CaM binding leads to CaMKII autophosphorylation at Thr286 and induces its translocation from the center of the dendritic spine to the post-synaptic density (PSD). Here, CaMKII co-localizes with NMDA receptors, positioning the holoenzyme in proximity to multiple substrates, such as AMPA receptors. (**B**) Ischemia leads to excitotoxicity concomitant with the excessive release of glutamate, overstimulation of glutamate receptors and intracellular Ca^2+^ overload. Increased Ca^2+^ influx rapidly activates CaMKII and leads to excessive autophosphorylation and translocation to the PSD.

**Figure 4 brainsci-12-01639-f004:**
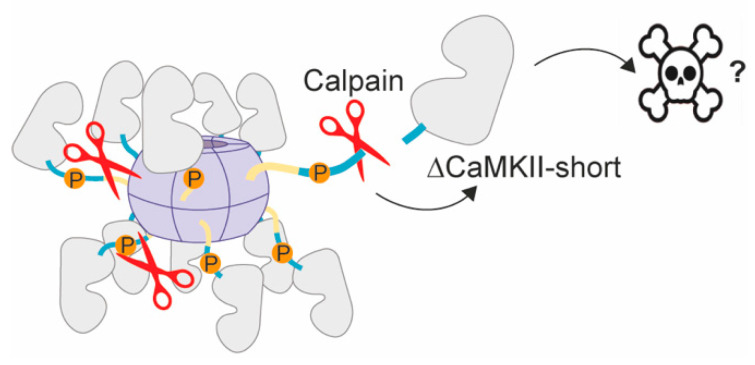
CaMKII cleavage by calpain after ischemia [113]. Ischemic conditions lead to overactivation of calpains (red scissors) as well as increased Thr286/287 CaMKII autophosphorylation, facilitating cleavage of CaMKII by overactivated calpains at M^281^↓H^282^. This leads to the ischemia-specific expression of ΔCaMKII consisting of the kinase domain and the N-terminal end of the regulatory domain, which is devoid of the Thr286/287 and CaM-binding region responsible for autoinhibition of the kinase.

**Figure 5 brainsci-12-01639-f005:**
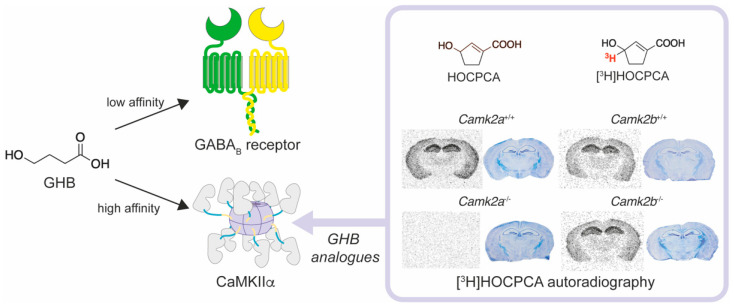
CaMKIIα as high-affinity binding site for γ-hydroxybutyrate (GHB) [25,148]. GHB targets both high- and low-affinity binding sites within the CNS. One well-established low-affinity binding site is the GABA_B_ receptor, and the high-affinity binding site was recently identified as CaMKIIα. *GHB analogues* display improved affinity and selectivity for CaMKIIα compared to GHB. E.g., the GHB analogue 3-hydroxycyclopent-1-enecarboxylic acid (HOCPCA) does not target GABA_B_ receptors and its selectivity was studied by showing the complete absence of [^3^H]HOCPCA binding to tissue from *Camk2a^−/−^* mice. In contrast, binding to *Camk2a^+/+^*, *Camk2b^−/−^* and *Camk2b^+/+^* mouse brain tissue was preserved.

**Table 1 brainsci-12-01639-t001:** Summary of in vivo neuroprotective effects of the Tat peptide CN21 and CN19o as well as the GHB analogue 3-hydroxycyclopent-1-enecarboxylic acid (HOCPCA). Of note, CN-peptides are administered intravenously and HOCPCA intraperitoneally [24,25,26,145].

	CN-Peptides	HOCPCA
In vivo Ischemic stroke	1 mg/kg tat-CN21 at 1 h post tMCAO ^1^ improves infarct volume in young male mice [24]	175 mg/kg HOCPCA at 30 min, 3 h, 6, h and 12 h and 90 mg/kg HOCPCA at 3 h post photothrombotic stroke improves infarct size and motor function (cylinder and grid walking task) in young male mice [25] 175 mg/kg HOCPCA at 3 h post photothrombotic stroke improves infarct size and motor function (cylinder and grid walking task) in aged female mice [25] 175 mg/kg HOCPCA at 30 min post pMCAO ^1^ improves infarct size and motor function (grip strength) in young male mice [145] 175 mg/kg HOCPCA at 3 h post pMCAO improves motor function (grip strength) in young male mice [145] 175 mg/kg HOCPCA at 30 min post thromboembolic stroke improves motor function (grip strength) in young male mice [145]
In vivo Global ischemia	Intracerebroventricular injection of 50 mg tat-CN21 at 3 h after global ischemia improves neuronal survival in the hippocampus and improved memory function (Barnes maze) [146] 0.01 mg/kg, 0.1 mg/kg and 1 mg/kg of tat-CN19o at 30 min post cardiac arrest and cardiopulmonary resuscitation (CA/CPR) improves neuronal survival in the hippocampus and improved memory function (contextual fear conditioning) [26]	

^1^ tMCAO: transient middle cerebral artery occlusion; pMCAO: permanent middle cerebral artery occlusion.

## Data Availability

No new data were created or analyzed in this study. Data sharing is not applicable to this article.

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
