# Peer review of "CaMKIIα as a Promising Drug Target for Ischemic Grey Matter"

_brainsci, 2022, doi:10.3390/brainsci12121639_

Round 1
Reviewer 1 Report
This review discusses the role of CaMKII in ischemic stroke and how inhibition of CaMKII could be a potential treatment. Overall, the review is clear and well-written, with figures that help explain the text. The authors could smooth out the writing in places, some of which I’ve noted below.
Text edits
-Line 79: isoforms is the wrong word here, I suggest variants or paralogs
-it’s redundant to say ‘hub association’ – cut it down to simply ‘hub’
-you could also cite the recent structure of CaMKII kinase domain bound to NMDA peptide (PMID: 35830796) line 207 or elsewhere
-line 282 should be ‘fail’
-line 322 should be ‘are’
-line 323 – does ‘excessive’ activation mean prolonged activation? Or that normally not all CaMKII holoenzymes are activated during a stimulus? It would be helpful to detail what is meant here. It’s also unclear what is meant by ‘excessive’ phosphorylation at Thr286, since the residue can only be phosphorylated once, perhaps you mean that more holoenzymes get phosphorylated? Or the phosphorylation lasts longer than normal?
-line 344 should be reworded from ‘could for instance be shown’ since it has already been shown this doesn’t make sense
-line 396 should change ‘could be’ to has been
-line 426 change ‘could also be’
-would be good to split up the paragraph starting at line 414
Author Response
This review discusses the role of CaMKII in ischemic stroke and how inhibition of CaMKII could be a potential treatment. Overall, the review is clear and well-written, with figures that help explain the text. The authors could smooth out the writing in places, some of which I’ve noted below.
Author comments:
Thank you for the great suggestions and comments. We have amended as detailed below.
-Line 79: isoforms is the wrong word here, I suggest variants or paralogs
We have adjusted the text to variants.
-it’s redundant to say ‘hub association’ – cut it down to simply ‘hub’
We have adjusted to ‘hub’ in line 76 and 131.
-you could also cite the recent structure of CaMKII kinase domain bound to NMDA peptide (PMID: 35830796) line 207 or elsewhere
We have added this citation in line 210.
-line 282 should be ‘fail’
We have corrected this.
-line 322 should be ‘are’
We have corrected this.
-line 323 – does ‘excessive’ activation mean prolonged activation? Or that normally not all CaMKII holoenzymes are activated during a stimulus? It would be helpful to detail what is meant here. It’s also unclear what is meant by ‘excessive’ phosphorylation at Thr286, since the residue can only be phosphorylated once, perhaps you mean that more holoenzymes get phosphorylated? Or the phosphorylation lasts longer than normal?
Yes, with excessive activation and autophosphorylation we meant to say that CaMKII undergoes prolonged activation, which was for example shown both by increased and sustained Thr286 autophosphorylation after ischemic stroke in mice in comparison to naïve wildtype mice. We understand that the wording was unclear, and we have rephrased the sentence to avoid the use of ‘excessive’:
Several mechanisms for its dysregulation have been described, and the majority of studies are in agreement with the following overall sequence of events. Initially, intracellular Ca2+ overload introduced by excitotoxic glutamate signaling results in rapid activation of CaMKII via Ca2+/CaM binding. Subsequently, CaMKII undergoes increased and sustained autophosphorylation at Thr286 after ischemia in comparison to sham non-stroke controls, rendering CaMKII autonomously active. Moreover, CaMKII gets persistently translocated to the PSD, where it co-localizes with the NMDA receptor subunit GluN2B.
-line 344 should be reworded from ‘could for instance be shown’ since it has already been shown this doesn’t make sense
This has been reworded to ‘has for instance been shown’.
-line 396 should change ‘could be’ to has been
We have corrected this.
-line 426 change ‘could also be’
We have changed this to ‘has been’
-would be good to split up the paragraph starting at line 414
We have split the paragraph.
Reviewer 2 Report
Tremendous efforts have been made to develop therapies targeting underlying cell death mechanisms to save dying tissue acutely after the initial injury (neuroprotectants), or therapies aiming to augment the function of surviving tissue and promote functional recovery after stroke. Failures in the design of both preclinical and clinical studies have, however, among other reasons, hampered translation into the clinic. However, CaMKII is reported to be dysregulated after ischemia in several previous reports, the authors introduce understandings of how ischemia affects CaMKII, and a possibility of CaMKII inhibition as a target for ischemic stroke.
I am afraid that the review is not clinically sound at all. As the authors mentioned, numerous neuroprotective agents have failed to demonstrate positive effects in the translational research, while they successfully showed therapeutic effects such as reduced infarcted volume, improved neurologic deficit, and reduced ROS using rodents in basic research. However, the authors described that CaMKII emerged as a promising target for neuroprotective treatments for ischemic stroke and potentially other diseases characterized by glutamate and Ca2+ dysregulation. The authors did not take into consideration the difference between grey matter and white matter to begin with. NMDA receptors and CaMKII are located in only postsynaptic region, or only grey matter. Critically, most of ischemic stroke occupy both grey and white matters in cardioembolic stroke, large-artery atherosclerosis, and embolic stroke of undetermined source, or only white matter in small vessel occlusion. Targeting only grey matter has been failing to demonstrate therapeutic effects in clinical trials, because pathomechanisms of ischemic stroke in grey matter are totally different from those in white matter mainly consisting of lipid hyperoxidization.
Thus, a review showing therapy targeting only gray matter is not significant.
Author Response
Reviewer 2
Tremendous efforts have been made to develop therapies targeting underlying cell death mechanisms to save dying tissue acutely after the initial injury (neuroprotectants), or therapies aiming to augment the function of surviving tissue and promote functional recovery after stroke. Failures in the design of both preclinical and clinical studies have, however, among other reasons, hampered translation into the clinic. However, CaMKII is reported to be dysregulated after ischemia in several previous reports, the authors introduce understandings of how ischemia affects CaMKII, and a possibility of CaMKII inhibition as a target for ischemic stroke.
I am afraid that the review is not clinically sound at all. As the authors mentioned, numerous neuroprotective agents have failed to demonstrate positive effects in the translational research, while they successfully showed therapeutic effects such as reduced infarcted volume, improved neurologic deficit, and reduced ROS using rodents in basic research. However, the authors described that CaMKII emerged as a promising target for neuroprotective treatments for ischemic stroke and potentially other diseases characterized by glutamate and Ca2+ dysregulation. The authors did not take into consideration the difference between grey matter and white matter to begin with. NMDA receptors and CaMKII are located in only postsynaptic region, or only grey matter. Critically, most of ischemic stroke occupy both grey and white matters in cardioembolic stroke, large-artery atherosclerosis, and embolic stroke of undetermined source, or only white matter in small vessel occlusion. Targeting only grey matter has been failing to demonstrate therapeutic effects in clinical trials, because pathomechanisms of ischemic stroke in grey matter are totally different from those in white matter mainly consisting of lipid hyperoxidization.
Thus, a review showing therapy targeting only gray matter is not significant.
Author comments:
Thank you for the comment about white matter injury which we have considered carefully:
We agree with the reviewer that pathomechanisms differ between ischemic injury to grey and white matter and that future development of neuroprotective therapies should ideally cover both pathologies. However, results from a recent Phase III trial (ESCAPE-NA1) set an important precedence that potential new neuroprotective treatments targeting grey matter pathology and utilizing animal models where the brain is largely made up of grey matter can produce positive translational data though into in a clinical setting. Specifically, the trial showed that targeting a postsynaptic pathway (inhibiting the interaction of NMDA receptors and PSD-95 with nerinetide) benefited a subgroup of patients that were not receiving alteplase. This highlights that NMDA receptor-mediated excitotoxic signaling is a clinically relevant target for neuroprotective drugs. As CaMKII is most known for its neuronal functions in decoding glutamate signaling directly downstream of NMDA receptors, pharmacological modulation of CaMKII and in particular the hub domain, which has received little to no attention until recently, is worth considering and discussing for the development of novel neuroprotective treatments based, especially on the back of the ESCAPE-NA1 trial. We do acknowledge that the contribution of CaMKII to ischemic cell death in other cell types of the neurovascular unit is less known and warrants further studies, however, this was not the focus of this review.
To make the focus of the review clearer, we have added the following in line 253:
Of note, this review focuses on neuronal CaMKII signaling in grey matter pathology. In contrast, the entire role of CaMKII as a target in white matter remains to be investigated.
Furthermore, we have rephrased the following paragraph of the conclusion (starting from line 645; changes highlighted in blue text):
Temporal and mechanistic delineation of the ischemia-induced dysregulation of CaMKII would be helpful to guide dosing regimens of pharmaceutical interventions targeting this important brain kinase. This would ideally include different types of ischemic injury (e.g., permanent versus gradual reperfusion and grey matter versus white matter injury models) and the combination with reperfusion therapy modelling thrombectomy or drug-induced thrombolysis [17]. Furthermore, more studies are needed to investigate the contribution of CaMKII signaling to ischemic cell death in non-neuronal cells such as endothelial cells or oligodendrocytes [163] or even infiltrating immune cells. Finally, with the eventual biological validation of targeting CaMKII in ischemic stroke, a broader range of acute brain injuries might be interesting to consider.
Reviewer 3 Report
The review paper entitled "CaMKII as a drug target for ischemic stroke" is very well structured and presented. The review highlights advances in the understanding of how ischemia affects CaMKII and how pathospecific pharmacological targeting of CaMKII signaling could be achieved.
In line 256 replace “surrounding the infract core ” by “surrounding the infarct core”
Author Response
The review paper entitled "CaMKII as a drug target for ischemic stroke" is very well structured and presented. The review highlights advances in the understanding of how ischemia affects CaMKII and how pathospecific pharmacological targeting of CaMKII signaling could be achieved.
Author comments:
Thank you for pointing out this typo. We have corrected:
In line 256 replace “surrounding the infract core ” by “surrounding the infarct core”
Round 2
Reviewer 2 Report
The authors responded that ESCAPE-NA1 set an important precedence that potential new protective treatments targeting grey matter pathology. However, the phase III trial eventually failed to show positive effects of nerinetide against ischemic stroke. Targeting only grey matter cannot be a novel treatment for ischemic stroke because clinically, most of ischemic stroke occupy both grey and white matters in cardioembolic stroke, large-artery atherosclerosis, and embolic stroke of undetermined source, or only white matter in small vessel occlusion as I commented previously. White matter pathologies must not be ignored.
Therefore, I am afraid I think the current title should be changed to more precise content ‘CaMKIIa as a promising drug target for grey matter ischemic change’, for example.
Please take into consideration my comment.
Author Response
Thank you for the constructive comment. We would like to suggest the following title:
"CaMKIIα as a promising drug target for ischemic grey matter"